# Nutrient Use Efficiency as a Strong Indicator of Nutritional Security and Builders of Soil Nutrient Status through Integrated Nutrient Management Technology in a Rice-Wheat System in Northwestern India

**Mehakpreet Kaur Randhawa [1], Salwinder Singh Dhaliwal [1,\*], Vivek Sharma [1], Amardeep Singh Toor [1], Sandeep Sharma [1], Manpreet Kaur [2] and Gayatri Verma [3]**

[1] Department of Soil Science, Punjab Agricultural University, Ludhiana 141004, India; mehakkaur15@gmail.com (M.K.R.); sharmavivek@pau.edu (V.S.); amardeep@pau.edu (A.S.T.); sandyagro@pau.edu (S.S.)

[2] Department of Chemistry, Punjab Agricultural University, Ludhiana 141004, India; manpreetchem@pau.edu

[3] Regional Research Station, Punjab Agricultural University, Gurdaspur 143521, India; drgayatriverma@pau.edu

\* Correspondence: ssdhaliwal@pau.edu

**Abstract:** Nutrient use efficiency is reported as a strong indicator of the buildup soil nutrient status for nutritional security of crops through an integrated nutrient management approach under a rice-wheat system. The data revealed that integrated application of manures and fertilizers reported maximum organic carbon (0.39%) in the treatment receiving 100% of the recommended dose of fertilizers (RDF) + farmyard manure and lowering the pH to 6.39. The maximum available N (360.8 kg ha$^{-1}$) was found in 100% RDF + press mud treatment; available P (66.30 kg ha$^{-1}$) was found in 75% RDF + poultry manure; and available K, Zn, Cu, and Fe (226.3 kg ha$^{-1}$ and 2.220, 0.732, and 36.87 mg kg$^{-1}$, respectively) in 100% RDF + farmyard manure treatments. Similarly, total macro- and micronutrient content in soil increased with the addition of organic manures alone or in combination with chemical fertilizers. The highest agronomic efficiency and utilization efficiency of nitrogen (41.83 and 102.55 kg kg$^{-1}$, respectively) and phosphorous (83.57 and 204.9 kg kg$^{-1}$, respectively) were recorded in the treatment receiving 75% RDF + poultry manure. This study concluded that the integrated application of manures and chemical fertilizers is a must for improving soil nutrient status and nutrient use efficiency and ultimately enhances nutritional security under a rice-wheat system.

**Keywords:** physicochemical properties; macronutrients; micronutrients; nutrient-use efficiency; rice-wheat system

## 1. Introduction

The rapid rate of increase in global population is exerting pressure on land resources due to intensive cultivation practices, which are ultimately degrading the soil. There is an urgent need to maintain soil health and quality for sustainable land production [1]. The use of chemical fertilizers is one of the ways to restore depleted nutrients. However, the higher cost of chemical fertilizers coupled with their low affordability by small farmers is the biggest obstacle for their use [2]. Additionally, the improper and imbalanced use of chemical fertilizers deteriorated soil health by producing serious effects on soil microorganisms, soil enzymatic activities, the atmosphere, and groundwater quality, causing health hazards and climate change [3]. It has also declined nutrient-use efficiency, making the consumption of chemical fertilizers uneconomical.

Nutrient use efficiency is an important concept for the evaluation of a crop production system. The aim of nutrient use efficiency is to improve the performance of crops by

providing economically optimal nourishment for crops and by minimizing the loss of nutrients from the field [4], and it is greatly affected by fertilizer, soil, and plant management practices. Earlier, to meet the major challenges in agriculture, intensive cropping was adopted, but now, the declining response to inputs has emerged as a major challenge for the sustainability of rice-wheat systems [5]. It is estimated that about 30–50% of nitrogen and 45% of phosphorus applied through fertilizers are used for crops [6]. The efficiency of nutrients can be increased through the integrated use of organic manures and chemical fertilizers [7].

The basic concept underlying integrated nutrient management is maintenance of the fertility of soil, sustainment of crop productivity, and improvement in farmer's profit through judicious and efficient use of chemical fertilizers, organic manures, and biofertilizers [8]. The organic manures have been used by farmers from ancient times. They increase the efficiency of applied nutrients in a rice-wheat system [9]. They not only supply nutrients to the current crop but also leave residual effects on the succeeding crop. The use of organic manures has beneficial effects on soil health by improving the physicochemical properties of soil besides supplying macronutrients and increasing the availability of micronutrients [10]. Organic manures also improve soil chemical properties such as supply and retention of soil nutrients and promotion of favorable enzymatic reactions [11]. Among various manures, farmyard manure, poultry manure, press mud, and rice straw compost are used in the present study. Farmyard manure is a reservoir of nutrients and is known to improve soil productivity on a sustainable basis [12]. Press mud is also an important organic wastes and is a good source of organic matter, macronutrients, as well as important micronutrients [13]. Poultry manure is applied in calcareous soils for the cultivation of crops due to a high concentration of macronutrients and micronutrients [14]. The burning of rice residues is emerging as a major issue. Though it is a cost-effective method of rice straw disposal, it causes a serious problem of environmental pollution, decreased organic matter content, losses in nutrients from soil, and harm to the microbial population residing in soil [15]. Rice straw compost (RSC) is one of the solutions to the problem of straw disposal [16].

It has been reported through various long-term studies that neither chemical fertilizers nor organic manures alone can achieve sustainability in crop production. The integrated use of manures and fertilizers assumes a great importance as manures not only improve soil productivity and sustainability but also reduce the mounting pressure on inorganic fertilizers [17]. Therefore, the objectives of this study are to assess the effects of manures and fertilizers on the nutrient status of soil and nutrient use efficiency under a rice-wheat system.

## 2. Materials and Methods

### 2.1. Experimental Site

The field experiment was conducted during 2019–2020 on a rice-wheat system at the research farm of the Department of Soil Science, Punjab Agricultural University, and Ludhiana. The experimental field lies in the central zone of Punjab situated at 30°56′ N latitude and 75°52′ E longitude, with a mean elevation of 247 m above mean sea level. The climate of the area (Ludhiana) is a subtropical, semiarid type with hot and dry summers from April to June, followed by humid and sultry climates from June to September and chilly winters from November to January. The average rainfall of the area is 759 mm.

### 2.2. Treatment Description

The experiment under study included fifteen treatments involving different combinations of chemical fertilizers and organic manures (Table 1). All treatments were replicated thrice in a randomized block design. The experiment included two crops: rice and wheat. Nitrogen through chemical fertilizers was applied in three equal splits at 7, 21, and 42 days after transplanting and all of the phosphorus was applied in soil at the time of transplanting the rice. To the wheat crop, N was applied at first and second irrigations in two split

doses and all of the phosphorus was drilled at the time of sowing. Organic manures were mixed in the soil before transplanting the rice and these manures were farmyard manure (15 t ha$^{-1}$), poultry manure (6 t ha$^{-1}$), press mud (15 t ha$^{-1}$), and rice straw compost (6 t ha$^{-1}$). The chemical composition of organic manures is given in Table 2.

**Table 1.** Different combinations of organic manures and chemical fertilizers.

| Treatment No. | Treatment Details |
|---|---|
| T1 | Control |
| T2 | Farmyard manure |
| T3 | Poultry manure |
| T4 | Press mud |
| T5 | Rice straw compost |
| T6 | 75% RDF |
| T7 | 75% RDF * + farmyard manure |
| T8 | 75% RDF + poultry manure |
| T9 | 75% RDF + press mud |
| T10 | 75% RDF + rice straw compost |
| T11 | 100% RDF |
| T12 | 100% RDF + farmyard manure |
| T13 | 100% RDF + poultry manure |
| T14 | 100% RDF + press mud |
| T15 | 100% RDF + rice straw compost |

* RDF–recommended dose of fertilizers; 75% RDF—78.7 kg N ha$^{-1}$ and 22.5 kg P$_2$O$_5$ ha$^{-1}$ to rice crop, and 105 kg N ha$^{-1}$ and 30 kg P$_2$O$_5$ ha$^{-1}$ to wheat crop; 100% RDF—93.7 kg N ha$^{-1}$ and 46.9 kg P$_2$O$_5$ ha$^{-1}$ to rice crop, and 125 kg N ha$^{-1}$ and 62.5 kg P$_2$O$_5$ ha$^{-1}$ to wheat crop.

**Table 2.** Chemical composition of organic manures.

| Organic Manure | N | P | K | Zn | Cu | Fe | Mn |
|---|---|---|---|---|---|---|---|
| | % | | | mg kg$^{-1}$ | | | |
| Farmyard manure | 1.93 | 0.76 | 2.10 | 116.1 | 64.3 | 648.7 | 148.9 |
| Poultry manure | 2.18 | 1.79 | 1.63 | 136.3 | 56.7 | 521.5 | 432.0 |
| Press mud | 3.19 | 0.82 | 0.65 | 143.8 | 94.7 | 485.6 | 294.5 |
| Rice straw compost | 1.71 | 0.38 | 1.62 | 133.4 | 51.4 | 521.2 | 202.9 |

### 2.3. Collection of Soil Samples

The surface soil samples were collected before transplanting the rice in June 2019 and after harvesting the wheat in April 2020. From each plot, five samples were collected and mixed to obtain a representative sample of soil. The samples were air-dried, grounded in a wooden mortar, and sieved using a 2-mm plastic sieve.

### 2.4. Analysis of Soil Samples

The pH of the soil samples was determined using soil–water suspension in a 1:2 ratio [18], and electrical conductivity (EC) was determined using a soil–water supernatant in a 1:2 ratio [19]. The wet combustion method given by Walkley and Black [20] was used for determining the organic carbon (OC) content. The available nitrogen was determined using the alkaline permanganate method [21]. The method given by Olsen et al. [22] was used to determine the available phosphorus in soil. The available potassium in soil was determined using the method given by Merwin and Peech [23] in which soil is extracted using neutral ammonium acetate solution. For determining diethylene triamine pentaacetic acid (DTPA) extractable micronutrients in soil, the diethylene triamine pentaacetic acid (DTPA) method developed by Lindsay and Norvell [24] was used and measured on an atomic absorption spectrophotometer (Varian Model AAS FS 240). Total N was analyzed by the method of Page et al. [25] in which soil was digested with concentrated H$_2$SO$_4$ in the presence of Hibbard's mixture and determined by the micro-Kjeldahl method. For the determination of total P, K, and micronutrients, soil samples were digested using

1 mL perchloric acid ($HClO_4$), 5 mL hydrofluoric acid (HF), and 5–6 drops of nitric acid ($HNO_3$) according to the method given by Page et al. [25]. Total P was determined on a spectrophotometer using the molybdenum-blue method; total K was determined on a flame photometer; and total Zn, Cu, Fe, and Mn contents were determined using atomic absorption spectrophotometer.

### 2.5. Nutrient Use Efficiency

The estimated values of agronomic efficiency (AE), physiological efficiency (PE), Agro-physiological efficiency (APE), apparent recovery efficiency (ARE), and utilization efficiency (UE) were calculated using the following formulae [26].

$$\text{Agronomic efficiency (AE)} = (G_f - G_u)/N_a = \text{kg kg}^{-1}$$

$$\text{Physiological efficiency (PE)} = (Y_f - Y_u)/(N_f - N_u) = \text{kg kg}^{-1}$$

$$\text{Agro-physiological efficiency (APE)} = (G_f - G_u)/(N_f - N_u) = \text{kg kg}^{-1}$$

$$\text{Apparent recovery efficiency (ARE)} = (N_f - N_u) \times 100/N_a = \%$$

$$\text{Utilization efficiency (EU)} = \text{PE} \times \text{ARE} = \text{kg kg}^{-1}$$

where

$G_f$ is the grain yield in the fertilized plot (kg ha$^{-1}$),
$G_u$ is the grain yield in the control (kg ha$^{-1}$),
$Y_f$ is the biological yield of the fertilized plot (kg ha$^{-1}$),
$Y_u$ is the biological yield of the control (kg ha$^{-1}$),
$N_a$ is the quantity of nutrients applied (kg ha$^{-1}$),
$N_f$ is the nutrient uptake in the fertilized plot (kg ha$^{-1}$), and
$N_u$ is the nutrient uptake in the control (kg ha$^{-1}$).

### 2.6. Statistical Analysis

The experimental data were subjected to statistical analysis using randomized block design in statistical package SAS software. The treatments were compared using least significant difference at the 5% level of significance. Duncan's multiple range test (DMRT) was employed to assess the differences between treatment means.

## 3. Results

### 3.1. Impact of Integrated Nutrient Management Approach on Physicochemical Properties of Soil

The data mentioned in Table 3 revealed that the pH values in surface soil samples collected after harvesting the wheat ranged from 6.40 to 6.75 in all treatments. The residual effect of organic manures added before transplanting the rice decreased the soil pH compared to the control (T1), 75% RDF (T6), and 100% RDF (T11). The combined application of organic manures and fertilizers resulted in greater decrease in soil pH than the sole application of organic manures. However, the difference among different treatments was nonsignificant.

The data presented in Table 3 illustrated that the EC values varied from 0.13 to 0.22 dS m$^{-1}$. The effect of different treatments was nonsignificant on EC in soil. However, the higher values were reported with the application of manures either alone or in combination with 75% RDF and 100% RDF. The minimum EC was found in the control (0.13 dS m$^{-1}$), and the maximum was found in the 100% RDF + press mud treatment (0.22 dS m$^{-1}$).

**Table 3.** Effect of the integrated nutrient management approach on pH, electrical conductivity (EC), and organic carbon content in soil.

| Treatments | pH | EC (dS m$^{-1}$) | OC (%) |
|:---:|:---:|:---:|:---:|
| T1 | 6.74 [a] | 0.13 [a] | 0.13 [g] |
| T2 | 6.51 [a] | 0.15 [a] | 0.34 [bcd] |
| T3 | 6.61 [a] | 0.15 [a] | 0.28 [e] |
| T4 | 6.67 [a] | 0.16 [a] | 0.33 [bcd] |
| T5 | 6.62 [a] | 0.12 [a] | 0.30 [de] |
| T6 | 6.68 [a] | 0.15 [a] | 0.21 [f] |
| T7 | 6.47 [a] | 0.13 [a] | 0.35 [abcd] |
| T8 | 6.54 [a] | 0.14 [a] | 0.33 [bcde] |
| T9 | 6.42 [a] | 0.14 [a] | 0.33 [bcde] |
| T10 | 6.41 [a] | 0.14 [a] | 0.32 [cde] |
| T11 | 6.66 [a] | 0.14 [a] | 0.21 [f] |
| T12 | 6.39 [a] | 0.14 [a] | 0.39 [a] |
| T13 | 6.47 [a] | 0.19 [a] | 0.36 [abc] |
| T14 | 6.57 [a] | 0.22 [a] | 0.38 [ab] |
| T15 | 6.46 [a] | 0.18 [a] | 0.35 [abcd] |
| Initial status | 6.69 | 0.17 | 0.18 |
| LSD (0.05) | NS | NS | 0.044 |

Averaged values within a column, succeeded by different small letters ([a, b, c, d, e, f, g]) differ significantly between different treatments at $p < 0.05$ significance level.

The OC content ranged from 0.13% in the control to 0.39% in the 100% RDF + farmyard manure treatment (Table 3). The OC content was significantly higher in the treatments where organic manures were incorporated. Among different manures, a higher build up in OC was recorded with the application of farmyard manure and press mud followed by poultry manure and rice straw compost, which might be ascribed to differences in the nutrient composition of organic manures and their release pattern into the soil. The organic carbon content increased in all the treatments compared to their initial status (0.18%) except in the control (0.13%) in which no manure or fertilizer had been added.

### 3.2. Impact of Integrated Nutrient Management Approach on Buildup of Nutrient Status of Soil

3.2.1. Available Macronutrients (N, P, and K)

The available N content ranged from 185.1 kg ha$^{-1}$ in the control to 360.8 kg ha$^{-1}$ in the 100% RDF + press mud treatment (Table 4). A significant increase in available N was observed when manure was added alone or in combination with 75% RDF or 100% RDF compared to the control. However, the available N content in treatments T2 (farmyard manure), T3 (poultry manure), T4 (press mud), and T5 (rice straw compost) in which only manure had been added was less than T6 (75% RDF) and T11 (100% RDF). Among the different treatments, the application of press mud with 100% RDF registered the maximum available N content, which was 94.92, 39.25, and 41.49% higher than T1 (control), T6 (75% RDF), and T11 (100% RDF), respectively.

The available P content of the soil ranged from 20.67 kg ha$^{-1}$ in the control to 66.73 kg ha$^{-1}$ in the 75% RDF + poultry manure treatment (Table 4). All the treatments that received organic manures alone or in combination with chemical fertilizers reported a significant buildup of available P in the soil. The 75% RDF + poultry manure treatment resulted in maximum P content, which was 222.8% higher than the control, and it was statistically on par with the 100% RDF + poultry manure treatment, which resulted 172.5% higher than the control. Among the different manures, poultry manure resulted in higher available P content, and it was statistically on par with farmyard manure, which might have been due to the difference in their P content.

**Table 4.** Effect of the integrated nutrient management approach on available macro- and micro nutrients in soil.

| Treatments | Available Macronutrients (kg ha$^{-1}$) | | | DTPA-Extractable Micronutrients (mg kg$^{-1}$) | | | |
|---|---|---|---|---|---|---|---|
| | N | P | K | Zn | Cu | Fe | Mn |
| T1 | 185.1 [j] | 20.67 [g] | 148.8 [e] | 0.772 [d] | 0.390 [j] | 16.32 [fd] | 2.983 [m] |
| T2 | 251.6 [g] | 41.46 [de] | 218.4 [ab] | 1.829 [abc] | 0.665 [c] | 26.97 [abcd] | 4.446 [f] |
| T3 | 256.9 [f] | 39.27 [de] | 200.0 [bc] | 1.302 [bcd] | 0.599 [e] | 28.49 [abcd] | 3.401 [l] |
| T4 | 237.7 [h] | 35.45 [def] | 200.2 [bc] | 1.730 [abc] | 0.656 [c] | 21.91 [bcd] | 5.491 [a] |
| T5 | 229.4 [i] | 25.42 [fg] | 206.8 [bc] | 1.461 [abcd] | 0.532 [h] | 24.66 [abcd] | 4.152 [i] |
| T6 | 259.1 [f] | 24.33 [fg] | 163.1 [de] | 0.907 [d] | 0.399 [j] | 17.73 [cd] | 3.515 [k] |
| T7 | 333.1 [d] | 43.46 [cd] | 225.4 [a] | 2.059 [ab] | 0.703 [b] | 32.67 [ab] | 4.332 [g] |
| T8 | 329.2 [d] | 66.73 [a] | 196.7 [c] | 2.101 [ab] | 0.580 [f] | 28.93 [abcd] | 4.741 [e] |
| T9 | 333.1 [d] | 39.14 [de] | 207.1 [bc] | 1.889 [ab] | 0.656 [c] | 26.20 [abcd] | 4.931 [d] |
| T10 | 266.4 [e] | 29.20 [efg] | 205.0 [bc] | 1.828 [abc] | 0.551 [g] | 25.73 [abcd] | 4.256 [h] |
| T11 | 255.0 [fg] | 29.90 [efg] | 167.3 [d] | 1.020 [cd] | 0.409 [i] | 18.52 [cd] | 4.152 [i] |
| T12 | 340.4 [c] | 54.02 [bc] | 226.3 [a] | 2.220 [a] | 0.732 [a] | 36.87 [a] | 5.016 [c] |
| T13 | 350.6 [b] | 56.34 [ab] | 217.3 [ab] | 1.732 [abc] | 0.618 [d] | 31.30 [abc] | 3.895 [j] |
| T14 | 360.8 [a] | 43.73 [cd] | 202.1 [bc] | 1.803 [abc] | 0.656 [c] | 27.45 [abcd] | 5.130 [b] |
| T15 | 352.8 [b] | 31.44 [defg] | 203.6 [bc] | 2.063 [ab] | 0.542 [g] | 26.31 [abcd] | 4.399 [f] |
| Initial status | 150.0 | 24.40 | 133.0 | 0.480 | 0.260 | 1.400 | 4.270 |
| LSD (0.05) | 4.886 | 11.29 | 16.16 | 0.705 | 0.010 | 11.68 | 0.061 |

Averaged values within a column, succeeded by different small letters ([a, b, c, d, e, f, g, h, i, j, k, l, m]) differ significantly between different treatments at *p* < 0.05 significance level.

The available K content ranged from 148.8 kg ha$^{-1}$ in the control to 226.3 kg ha$^{-1}$ in the 100% RDF + farmyard manure treatment (Table 4). The application of different manures either alone or in combination with chemical fertilizers resulted in a higher buildup of available K in soil compared to the control and the sole applications of 75% RDF and 100% RDF. The proportionate increase in available K content was more with the farmyard manure compared to other manures, which could have been due to the variation in K content in the different manures. Among the different treatments, the 100% RDF + farmyard manure resulted in maximum available K content in soil, which was statistically on par with the 75% RDF + farmyard manure, the farmyard manure, and the 100% RDF + poultry manure treatments and increased the available K content by 52.08, 51.47, 46.77, and 46.03%, respectively, over the control.

3.2.2. Total Macronutrients (N, P, and K)

The data presented in Table 5 shows the significant influence of manures and fertilizers on total N content in soil, which ranged from 0.056 to 0.133% in all the treatments. The maximum total N content was found in the 100% RDF + poultry manure and the 100% RDF + press mud treatments, which were significantly different from all other treatments and 137.5% higher than the control. The application of different manures in combination with 75% RDF and 100% RDF resulted in higher total N content than the sole use of different manures and chemical fertilizers (T2 (farmyard manure), T3 (poultry manure), T4 (press mud) and T5 (rice straw compost), T6 (75% RDF), and T11 (100% RDF)). Among the different manures, poultry manure (0.080 to 0.133%) and press mud (0.076 to 0.133%) showed higher ranges than farmyard manure (0.070 to 0.106%) and press mud (0.066 to 0.093%) treatments. Among the different organic manures, press mud, poultry manure, and farmyard manure registered greater increases in the available N content of soil compared to rice straw compost.

**Table 5.** Effect of the integrated nutrient management approach on total macro- and micronutrients in soil.

| Treatments | Total Macronutrients (%) | | | Total Micronutrients (mg kg$^{-1}$) | | | |
|---|---|---|---|---|---|---|---|
| | N | P | K | Zn | Cu | Fe | Mn |
| T1 | 0.056 [i] | 0.032 [j] | 0.290 [k] | 28.05 [d] | 287.5 [a] | 12,333.3 [d] | 198.8 [g] |
| T2 | 0.070 [g] | 0.034 [i] | 0.710 [d] | 46.20 [bc] | 340.0 [a] | 20,552.5 [ab] | 339.5 [ab] |
| T3 | 0.080 [e] | 0.036 [g] | 0.537 [f] | 44.25 [bc] | 335.0 [a] | 18,475.0 [abc] | 302.5 [bcd] |
| T4 | 0.076 [f] | 0.040 [f] | 0.496 [h] | 46.18 [bc] | 281.7 [a] | 18,437.5 [abc] | 266.7 [de] |
| T5 | 0.066 [h] | 0.033 [ij] | 0.382 [j] | 41.45 [c] | 360.0 [a] | 17,580.0 [abc] | 288.4 [bcde] |
| T6 | 0.071 [g] | 0.035 [h] | 0.430 [i] | 33.35 [d] | 292.5 [a] | 13,750.0 [cd] | 206.3 [fg] |
| T7 | 0.085 [c] | 0.044 [d] | 0.855 [a] | 45.78 [bc] | 332.5 [a] | 21,645.0 [a] | 357.1 [a] |
| T8 | 0.093 [c] | 0.045 [c] | 0.696 [d] | 50.02 [ab] | 355.0 [a] | 18,370.0 [abc] | 312.5 [abcd] |
| T9 | 0.085 [d] | 0.047 [b] | 0.494 [h] | 47.80 [ab] | 312.5 [a] | 18,953.8 [ab] | 322.5 [abc] |
| T10 | 0.080 [e] | 0.040 [f] | 0.556 [e] | 42.02 [c] | 288.3 [a] | 17,682.5 [abc] | 250.0 [ef] |
| T11 | 0.085 [d] | 0.036 [g] | 0.515 [g] | 33.25 [d] | 287.5 [a] | 13,755.0 [cd] | 212.5 [fg] |
| T12 | 0.106 [b] | 0.047 [b] | 0.857 [a] | 49.65 [ab] | 361.7 [a] | 20,320.0 [ab] | 320.0 [abc] |
| T13 | 0.133 [a] | 0.045 [c] | 0.774 [c] | 45.10 [bc] | 355.0 [a] | 19,985.0 [ab] | 291.3 [bcde] |
| T14 | 0.133 [a] | 0.051 [a] | 0.808 [b] | 53.70 [a] | 320.0 [a] | 19,555.0 [ab] | 315.0 [abcd] |
| T15 | 0.093 [c] | 0.043 [e] | 0.703 [d] | 44.20 [bc] | 320.0 [a] | 16,047.3 [bcd] | 276.3 [cde] |
| Initial status | 0.060 | 0.028 | 0.310 | 40.20 | 289.0 | 15,615.0 | 265.0 |
| LSD (0.05) | 0.002 | 0.001 | 0.016 | 6.226 | NS | 4406.2 | 45.99 |

Averaged values within a column, succeeded by different small letters ([a, b, c, d, e, f, g, h, i, j, k]) differ significantly between different treatments at $p < 0.05$ significance level.

The total P content varied from 0.032% in the control to 0.051% in the 100% RDF + press mud treatments, as shown in Table 5. The higher content of total P was found with the combined application of manures and fertilizers. The maximum total P was found in the 100% RDF + press mud treatment, which was 59.37, 45.71, and 41.67% higher than the control, 75% RDF, and 100% RDF treatments. Among the different manures, the application of press mud resulted in a higher total P content either alone or with 75% and 100% RDF.

The data in Table 5 revealed that total K content ranged from 0.290 to 0.857% among different treatments. The higher content of total K varying from 0.710 to 0.857% was observed in treatments where farmyard manure was added (T2 (farmyard manure), T7 (75% RDF + farmyard manure), and T12 (100% RDF + farmyard manure)), and they were significantly different from all other treatments. Furthermore, the data revealed that a lower content of total K was found in the control (0.290%), 75% RDF (0.430%), and 100% RDF (0.515%) treatments compared to the combined application of 75% and 100% RDF with different manures.

### 3.2.3. DTPA-Extractable Micronutrients (Zn, Cu, Fe, and Mn)

The perusal of data indicated that DTPA-extractable Zn in different treatments varied from 0.772 mg kg$^{-1}$ to 2.220 mg kg$^{-1}$, as shown in Table 4. The maximum DTPA-extractable Zn was reported in the 100% RDF + farmyard manure (T1) treatment, which was statistically on par with press mud (T4), rice straw compost (T5), 75% RDF + farmyard manure (T7), 75% RDF + press mud (T9), 75% RDF + poultry manure (T8), 100% RDF + poultry manure (T13), 100% RDF + press mud (T14), and 100% RDF + rice straw compost (T15) treatments. The data further showed that Zn in treatments with only 75% or 100% RDF resulted in lower values compared to treatments in which manures had been added either alone or in combination with 75% or 100% RDF.

The DTPA-extractable Cu ranged from 0.390 mg kg$^{-1}$ to 0.732 mg kg$^{-1}$ in different treatments, and among all the treatments, it was reported higher in organic manures plots (either alone or in combination with chemical fertilizers) compared to the control, 75% RDF, and 100% RDF treatments (Table 4). The highest DTPA-extractable Cu content was reported in the 100% RDF + farmyard manure treatment, which was significantly different from all other treatments and resulted in 87.69% higher Cu content over the control. Among

the different organic manures, the proportionate increase in DTPA-extractable Cu was recorded with the application of farmyard manure followed by press mud, poultry manure, and then rice straw compost.

The data reported in Table 4 revealed that DTPA-extractable Fe varied from 16.32 to 36.87 mg kg$^{-1}$ in all the treatments. The maximum DTPA-extractable Fe resulted from 100% RDF + farmyard manure treatment, which was 125.9% higher than the control. The addition of manure resulted in higher DTPA-extractable Fe than 75% RDF and 100% RDF, and among different manures, farmyard manure (26.97 to 36.87 mg kg$^{-1}$) and poultry manure (28.49 to 31.30 mg kg$^{-1}$) resulted in higher DTPA-extractable Fe content compared to press mud (21.91 to 27.45 mg kg$^{-1}$) and rice straw compost (24.66 to 26.31 mg kg$^{-1}$).

The DTPA-extractable Mn content ranged from 2.983 to 5.491 mg kg$^{-1}$ in different treatments (Table 4). The higher values of DTPA-extractable Mn were reported with the addition of organic manures compared to treatments in which only chemical fertilizers had been added. Among the different treatments, press mud resulted in the maximum and 84.07% higher Mn content than the control. Among the different organic manures, the application of press mud in treatments T4 (press mud), T9 (75% RDF + press mud), and T14 (100% RDF + press mud) registered higher DTPA-extractable Mn compared to other manures, and it was followed by the application of farmyard manure.

### 3.2.4. Total Micronutrients (Zn, Cu, Fe, and Mn)

The data reported for total Zn indicated that total Zn increased in all the treatments over the control (Table 5). The concentration of total Zn ranged from 28.05 to 53.70 mg kg$^{-1}$. The highest Zn content was found in the 100% RDF + press mud treatment (53.70 mg kg$^{-1}$), which was statistically on par with 75% RDF + poultry manure (50.02 mg kg$^{-1}$), 75% RDF + press mud (47.80 mg kg$^{-1}$), and 100% RDF + farmyard manure (49.65 mg kg$^{-1}$) treatments. Compared to other manures, lower total Zn contents was found with rice straw compost treatments (T5, T10, and T15) and higher ranges were found with press mud treatments (T4, T9, and T14).

The results mentioned in Table 5 showed that the total Cu content varied from 287.5 to 361.7 mg kg$^{-1}$ under all the treatments. The highest content of total Cu was reported in the 100% RDF + farmyard manure treatment (361.7 mg kg$^{-1}$), which was closely followed by 75% RDF + poultry manure (355.0 mg kg$^{-1}$) and 100% RDF + poultry manure (355.0 mg kg$^{-1}$) treatments and was minimum under the control (287.5 mg kg$^{-1}$). There was a nonsignificant difference among the different treatments, but a higher content of total Cu was found with the application of manures compared to the application of 75% RDF and 100% RDF without manures.

The total Fe content increased with the application of manure alone or in combination with 75% RDF and 100% RDF, and it varied from 12,333.3 to 21,645.0 mg kg$^{-1}$ among the different treatments. A higher content of total Fe was registered with the application of farmyard manure, which was 21,645.0 mg kg$^{-1}$ in 75% RDF + farmyard manure, 20,552.5 mg kg$^{-1}$ in farmyard manure, and 20,320.0 mg kg$^{-1}$ in 100% RDF + farmyard manure treatments followed by poultry manure treatments, and was lowest with the addition of rice straw compost. The lower content was found in the control, 75% RDF, and 100% RDF treatments.

The total Mn content in soil varied from 198.8 to 357.1 mg kg$^{-1}$ in all the treatments. The maximum content was found in the 75% RDF + farmyard manure treatment and minimum under the control. Among the different manures, farmyard manure resulted in a higher total Mn content (320.0 to 357.1 mg kg$^{-1}$) followed by poultry manure and press mud treatments and was minimum under rice straw compost treatments. There was no significant difference between the 75% RDF and 100% RDF treatments and the control but the combined application of chemical fertilizers with organic manures increased the total Mn content.

*3.3. Impact of Integrated Nutrient Management Approach on Nutrient Use Efficiency*

3.3.1. N-Use Efficiency in Wheat

The different treatment combinations exerted significant impacts on agronomic efficiency (AE), physiological efficiency (PE), agro-physiological efficiency (APE), apparent recovery efficiency (ARE), and utilization efficiency of nitrogen in wheat crop, and they ranged from 16.67 to 41.83 kg kg$^{-1}$, from 92.18 to 103.67 kg kg$^{-1}$, from 31.87 to 41.93 kg kg$^{-1}$, from 51.93 to 106.06%, and from 49.11 to 102.55 kg kg$^{-1}$, respectively (Table 6). A higher value of AE was recorded in the treatment in which crop received 75% RDF + poultry manure (T8), and it was statistically on par with 75% RDF + farmyard manure (T7) and 100% RDF + farmyard manure (T12). These three treatments were significantly superior to the rest of the treatments. The results indicated that the integrated use of chemical fertilizers and organic manures was better at improving AE compared to the sole applications of 75% RDF and 100% RDF. PE and ARE were more in the treatment receiving 75% RDF + farmyard manure (T7), which was statistically on par with the 75% RDF + rice straw compost treatment (T10). The treatment receiving 75% RDF + poultry manure (T8) resulted in higher values of ARE and UE.

**Table 6.** Nitrogen use efficiency as affected by the integrated nutrient management approach in wheat.

| Treatments | AE (kg kg$^{-1}$) | PE (kg kg$^{-1}$) | APE (kg kg$^{-1}$) | ARE (%) | UE (kg kg$^{-1}$) |
|:---:|:---:|:---:|:---:|:---:|:---:|
| T1–T5 | - | - | - | - | - |
| T6 | 16.67 [e] | 94.22 [cd] | 31.87 [e] | 51.93 [e] | 49.11 [f] |
| T7 | 40.82 [a] | 103.67 [a] | 41.93 [a] | 96.84 [ab] | 100.47 [ab] |
| T8 | 41.83 [a] | 96.28 [bcd] | 39.30 [b] | 106.06 [a] | 102.55 [a] |
| T9 | 32.72 [bc] | 97.95 [bc] | 36.48 [c] | 89.62 [bc] | 87.90 [bcd] |
| T10 | 34.01 [bc] | 99.30 [ab] | 40.84 [ab] | 83.24 [c] | 82.78 [cd] |
| T11 | 24.58 [d] | 92.83 [d] | 35.04 [cd] | 70.11 [d] | 65.14 [e] |
| T12 | 37.55 [ab] | 93.20 [cd] | 36.58 [c] | 102.61 [a] | 95.71 [abc] |
| T13 | 33.89 [bc] | 83.25 [e] | 34.20 [cde] | 98.98 [ab] | 82.49 [cd] |
| T14 | 30.13 [c] | 86.43 [e] | 33.47 [de] | 90.23 [bc] | 78.16 [de] |
| T15 | 35.44 [bc] | 92.18 [d] | 36.30 [c] | 97.57 [ab] | 90.16 [abcd] |
| LSD (0.05) | 5.369 | 5.024 | 2.396 | 11.26 | 14.40 |

Averaged values within a column, succeeded by different small letters ([a, b, c, d, e, f]) differ significantly between different treatments at *p* < 0.05 significance level.

3.3.2. P-Use Efficiency in Wheat

The data presented in Table 7 show that AE, PE, APE, ARE, and UE for phosphorus in wheat crop ranged from 30.30 to 83.57 kg kg$^{-1}$, from 200.61 to 388.27 kg kg$^{-1}$, from 77.65 to 141.75 kg kg$^{-1}$, from 25.81 to 68.68%, and from 98.1 to 204.9 kg kg$^{-1}$, respectively, under different treatments. The 75% RDF + poultry manure (T8) resulted in a maximum value of AE, and it was statistically on par with 75% RDF + farmyard manure (T7) and 100% RDF + farmyard manure (T12) treatments. The PE and ARE were more in the treatments receiving 75% RDF and 100% RDF, respectively. The 100% RDF + press mud treatment (T14) resulted in maximum value of ARE, and it was statistically on par with 75% RDF + farmyard manure (T7) and 75% RDF + poultry manure (T8) treatments. The maximum UE was recorded with the application of the 75% RDF + poultry manure treatment (T8). Overall, the treatments receiving combined applications of manures and fertilizers resulted in higher values of AE, ARE, and UE while 75% RDF and 100% RDF resulted in higher values of PE and APE than other treatments.

**Table 7.** Phosphorus use efficiency as affected by the integrated nutrient management approach in wheat.

| Treatments | AE (kg kg$^{-1}$) | PE (kg kg$^{-1}$) | APE (kg kg$^{-1}$) | ARE (%) | UE (kg kg$^{-1}$) |
|---|---|---|---|---|---|
| T1–T5 | - | - | - | - | - |
| T6 | 33.30 [f] | 388.27 [a] | 131.69 [ab] | 25.81 [e] | 98.1 [f] |
| T7 | 81.55 [ab] | 302.47 [bc] | 122.61 [ab] | 66.23 [abc] | 200.7 [ab] |
| T8 | 83.57 [a] | 298.60 [c] | 121.77 [ab] | 68.68 [ab] | 204.9 [a] |
| T9 | 65.36 [cd] | 305.58 [bc] | 113.84 [b] | 57.40 [bcd] | 175.6 [bcd] |
| T10 | 67.94 [cd] | 314.51 [abc] | 129.1 [ab] | 53.48 [d] | 165.4 [cd] |
| T11 | 49.17 [e] | 377.58 [ab] | 141.75 [a] | 35.94 [e] | 130.3 [e] |
| T12 | 75.10 [abc] | 299.44 [c] | 117.43 [ab] | 64.29 [bcd] | 191.4 [abc] |
| T13 | 67.78 [cd] | 304.44 [bc] | 125.08 [ab] | 54.47 [d] | 164.9 [cd] |
| T14 | 60.27 [d] | 200.61 [d] | 77.65 [c] | 77.74 [a] | 156.3 [de] |
| T15 | 70.89 [bcd] | 329.80 [abc] | 129.57 [ab] | 55.33 [cd] | 180.3 [abcd] |
| LSD (0.05) | 10.73 | 76.96 | 25.97 | 11.71 | 28.78 |

Averaged values within a column, succeeded by different small letters ([a, b, c, d, e, f]) differ significantly between different treatments at $p < 0.05$ significance level.

### 3.4. Relationship between Soil Organic Carbon and Available Soil Nutrients

Using the observations recorded on organic carbon content and available soil nutrients, linear relationships between organic carbon and available soil nutrients were derived and are presented in Figure 1a for the available N, P, and K; in Figure 1b for DTPA-extractable Zn and Cu; and in Figure 1c for DTPA-extractable Fe and Mn. The available macronutrient and DTPA-extractable micronutrient contents increased with the increase in organic carbon content of soil with the addition of organic manures. All available nutrients were significantly correlated with organic carbon content affected by the addition of organic manures and chemical fertilizers in soil. A significant positive correlation was found among organic carbon and available N, P, and K ($R^2 = 0.57$, 0.43, and 0.85, respectively) and DTPA-extractable Zn, Cu, Fe, and Mn ($R^2 = 0.82$, 0.79, 0.71, and 0.55, respectively).

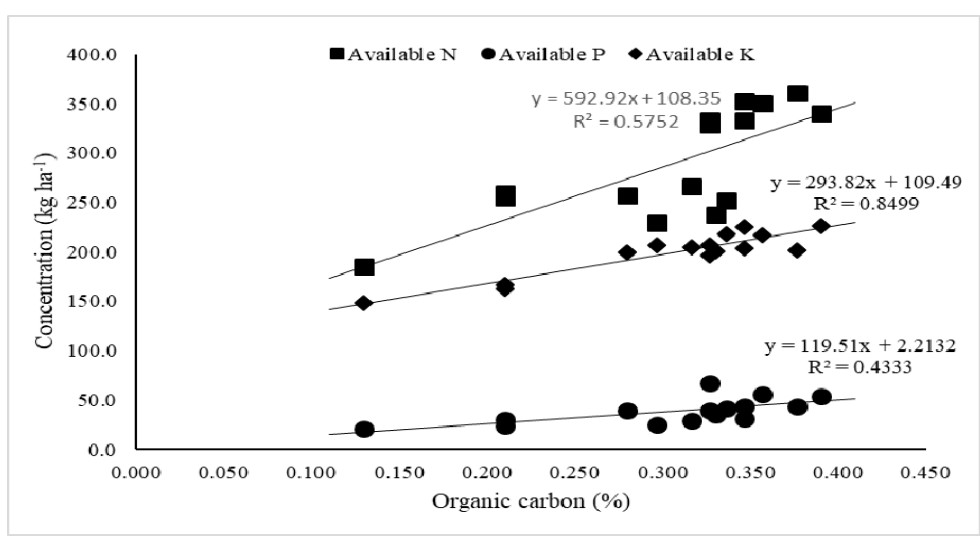

(**a**)

**Figure 1.** *Cont.*

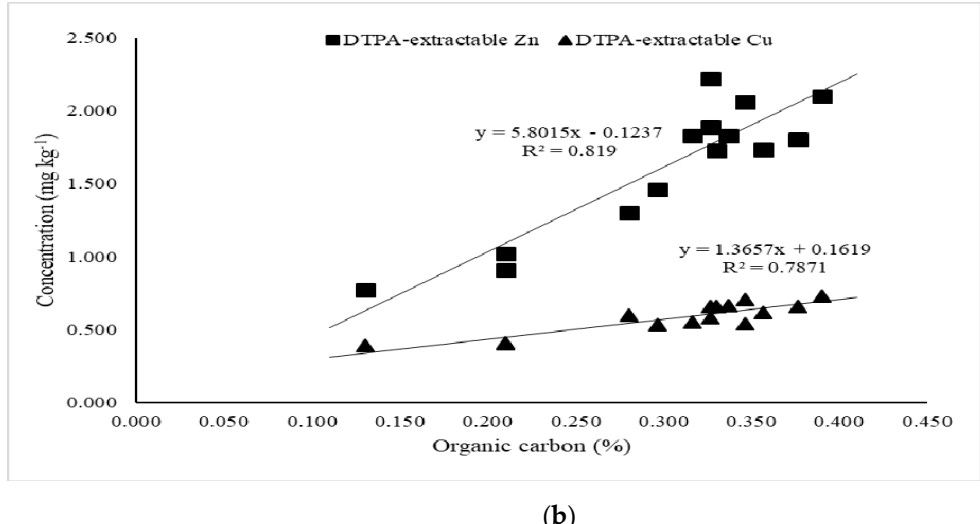

(**b**)

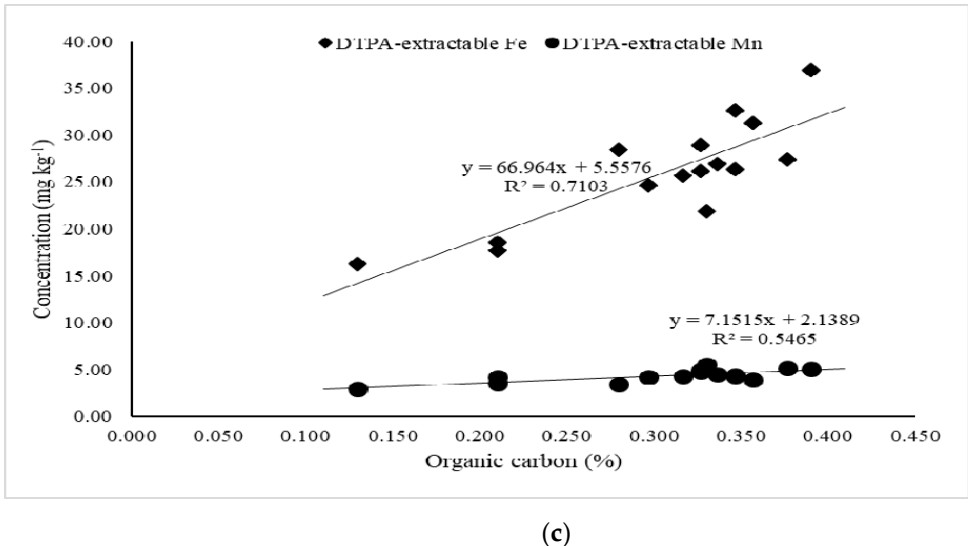

(**c**)

**Figure 1.** Relationship between organic carbon with available NPK (**a**), DTPA-Zn and DTPA-Cu (**b**) and DTPA-Fe and Mn (**c**)

## 4. Discussion

### 4.1. Impact of Integrated Nutrient Management Approach on Physicochemical Properties of Soil

The slight decline in pH in manured plots may be due to organic acids liberated during manure decomposition. The acid-producing nature of urea, which releases hydrogen ions during nitrification, could also have been responsible for decreasing soil pH [17]. The increase in EC with the addition of manures and fertilizers was due to the addition of soluble salts and the solubilization of native minerals due to a reduction in the pH of the soil [27]. The decomposition of organic manures released acids or acid-forming compounds that react with sparingly soluble salts already present in the soil and either converted them into soluble salts or at least increased their solubility [28].

The higher increase in OC content due to combined applications of manures and fertilizers may be because of better crop growth accompanied by higher root biomass generation and higher return of leftover plant residues [29]. The higher soil organic carbon in fertilized plots is due to the addition of carbon through roots and crop residues, the higher humification rate constant, and the lower decay rate [30]. The lower dry matter production and low return of crop residues and root biomass in the soil could have been responsible for the lower organic carbon content in the control, 75% NPK, and 100% NPK

treatments [1]. These results are consistent with the findings of Mishra et al. [31] and Sharma et al. [32].

### 4.2. Impact of Integrated Nutrient Management Approach on Buildup of Nutrient Status of Soil

4.2.1. Available Macronutrients (N, P, and K)

The direct addition of nitrogen through manures might have been the reason for the increase in available nitrogen in treatments in which manures and fertilizers were added. The results are confirm the findings of Hemalatha and Chellamuthu [33], Sharma [34], and Verde et al. [35]. It has been reported that the application of farmyard manure, poultry manure, press mud, and rice straw compost with 75% or 100% RDF resulted in higher available P contents compared to their sole applications. The higher availability of phosphorus in manure-amended treatments might have been attributed to the solubilization of P by organic acids released during the decomposition of organic manures, due to enhanced microbial activities, and due to a reduction in P fixation in soil with chelation of P fixing cations (Fe and Al) [36]. The mineralization of organic P present in manures also contributes to the available P in soil [1].

The higher content of available K in treatments in which organic manures were added could be due to the addition of K through manures that could supply a certain amount of K in soil through the interaction of organic matter with clay. Furthermore, the decomposition of manures leads to the release of organic colloids, which increases the cation exchange capacity of soil and can hold more exchangeable K in soil. The increased cation exchange capacity of soil also reduces leaching losses of K in soil. The decomposition of organic manures releases carbon dioxide, which dissolves in water to form carbonic acids, which is capable of decomposing certain primary minerals, leading to the release of nutrients [37]. The results are in agreement with the findings of Mazumdar et al. [38] and Sathish et al. [39].

4.2.2. Total Macronutrients (N, P, and K)

The increase in total nitrogen followed the increase in soil organic carbon content because the internal cycling in soil is linked to soil organic carbon. The increase in soil organic carbon and total nitrogen were observed by Kaur et al. [40] in organically managed soils compared to conventionally managed soils in which only chemical fertilizers were added. The addition of P in the treatments could be the main reason for the higher accumulation of total P in soil, and the addition of manures and fertilizers leads to large amounts of unutilized P [41]. The increase in total K in soils treated with manures could be attributed to the increase in available K and non-exchangeable K and due to the fixation of added soluble K through manures by clay minerals. The increase in total K with the addition of manures has also been reported by Kaur and Benipal [42], and Soremi et al. [43].

4.2.3. DTPA-Extractable Micronutrients (Zn, Cu, Fe, and Mn)

The addition of farmyard manure, poultry manure, press mud, and rice straw compost results in the formation of metal–organic complexes that increased the availability of micronutrients. The chelating action of organic manures has been found to increase the availability of micronutrients and aided in preventing their adsorption, fixation, and precipitation [44,45]. Dhaliwal et al. [10] reported a significant increase in DTPA-extractable micronutrients with manuring over the application of the recommended dose of fertilizers under a rice-wheat system. Parven et al. [46], Rutkowska et al. [47], and Singh et al. [48] also reported similar results. The addition of organic manures to soil also encourages microorganisms, which help in the liberation of available micronutrients. The increase in DTPA-extractable micronutrients with the application of organic manures is in agreement with the findings of Chauhan et al. [49] and Thakur et al. [50]. The regular addition of organic manures leads to an increase in organic carbon content, which increases the availability of micronutrients [51,52].

### 4.2.4. Total Micronutrients (Zn, Cu, Fe, and Mn)

The addition of different manures resulted in a buildup of total micronutrients in soil due to their higher release in soil. The increase in total Zn and Fe may be attributed to an increase in the redox-potential of soil with the addition of different manures, which leads to more added Zn and Fe in the total form [53]. Shahid et al. [54] reported an increase in the total content of Zn, Cu, Fe, and Mn with the application of the phosphorous fertilizer and farmyard manure either alone or in combination due to their micronutrient content and atmospheric depositions.

### *4.3. Impact of Integrated Nutrient Management Approach on Nutrient Use Efficiency*
### 4.3.1. N-Use Efficiency in Wheat

The application of manures played an important role in the mobilization of nutrients, leading to better availability of nutrients and facilitating uptake by plants, resulting in high nitrogen use efficiency [55]. The results indicate that all the treatments increased the production capacity per unit amount of nutrient applied, which might have been due to the prolonged availability of nitrogen [8].

### 4.3.2. P-Use Efficiency in Wheat

The results reported that physiological and agro-physiological efficiencies decreased with the addition of organic manures along with chemical fertilizers, which might be due to the crop needs being satisfied with the lower dose of nutrients, which were supplied without organic manures [56] The combined application of manures and fertilizers increased the agronomic efficiency, apparent recovery, and utilization efficiencies owing to the beneficial effects of the integrated use of manures and fertilizers, which had a positive influence on growth and yielded attributing characteristics. Additionally, the proper decomposition of manures increased the direct supply of nutrients to plants and created a favorable environment for the growth of plants [5].

### 5. Conclusions

From the present study, it may be concluded that the addition of organic manures in conjunction with chemical fertilizers improves the available as well as total nutrient status of soil compared to their individual application. Among the different manures, farmyard manure, poultry manure, and press mud were proven to be better than rice straw compost, which could have been due to the differences in their chemical composition and rate of decomposition in the soil. Apart from this, nitrogen and phosphorus use efficiency in wheat was enhanced with an increase in crop productivity, which is ultimately the result of improved soil conditions and uptake of available nutrients. Thus, the addition of organic manures with chemical fertilizers is a must for improving soil health and productivity and is the need of the hour in the present scenario.

**Author Contributions:** Writing-original draft preparation, M.K.R. and S.S.D.; Conceptualization, S.S.D., S.S., V.S and M.K.; Resources, A.S.T.; Supervision, S.S.D., V.S.; Software, G.V. All authors have read and agreed to the published version of the manuscript.

**Funding:** This research received no external funding.

**Institutional Review Board Statement:** Not applicable.

**Informed Consent Statement:** Not applicable.

**Data Availability Statement:** The data will be provided as per demand.

**Acknowledgments:** The authors sincerely thank the Indian Council for Agricultural Research, Indian Institute of Soil Science, Bhopal, for their financial assistance. We are also thankful to the head of the Department of Soil Science and administration, Punjab Agricultural University, Ludhiana for providing the necessary facility and technical guidance. The views expressed in this paper are those of individual scientists and do not necessarily reflect the views of the donor or the authors' institution.

**Conflicts of Interest:** The authors declare no conflict of interest.

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
