# Peer review of "Nutrient Use Efficiency as a Strong Indicator of Nutritional Security and Builders of Soil Nutrient Status through Integrated Nutrient Management Technology in a Rice-Wheat System in Northwestern India"

_sustainability, doi:10.3390/su13084551_

Round 1

Reviewer 1 Report

The goals and data of the work are interesting, but it is necessary to make some improvements, mainly in the Materials and Methods.

Some indications:

. The title is too long

. Appropriate introduction and objectives

Material and Methods

2.1. Experimental site and soil characteristics - Despite the title, this item makes no reference to the soil characteristics

. You should make a brief characterization of the soils, climate and geology of the experimental area

. It is necessary to make a clearer description of the treatments. What does the acronym RDF mean? Why did you choose these treatments? What factors were behind your choice?

. What is the size of the treatment plots? Is there a replication of treatments?

. What depth were the soil samples taken?

. Did you only perform the laboratory analysis of a soil sample per treatment (a mixture of 5 samples)? You only have one value per treatment! How is it possible to apply the mean comparison test? You should explain the methodological procedures very well!

Line 97-98 - “From each plot 5 samples were collected and they were mixed to get a representative sample of soil” - In this case, does Plot meaning treatment?

2.5. Nutrient use efficiency

Explain the information obtained from the parameters evaluated in this item. For what purpose was each of them evaluated?

Results

.Table 1 - The effects of treatments on soil pH and EC, despite undergoing small variations, are not significant, so there is not much to say. The OC increased significantly, but the values remain very low. Despite this, they bring benefits to the availability of nutrients for the plants. This fact should be highlighted in the texto.

. The results presentation and discussion, in my opinion, are adequate

Conclusion

You should improve the conclusion

Author Response

Comment 1: The title is too long.

Response 1: The title is already modified

Material and Methods

Comment 2: 2.1. Experimental site and soil characteristics - Despite the title, this item makes no reference to the soil characteristics.

Response 2: 2.1. Experimental site and soil characteristics is changed to Experimental site.

Comment 3: You should make a brief characterization of the soils, climate and geology of the experimental area.

Response 3: All the characteristics of experimental soil are given in Table 3, 4 and 5 as initial status and some additions are made in section 2.1 giving a brief characterization of climate and geology of the area.

Comment 4 : It is necessary to make a clearer description of the treatments. What does the acronym RDF mean? Why did you choose these treatments? What factors were behind your choice? .

Response 4: The acronym RDF means recommended dose of fertilizers. The reasons behind choosing these treatments are already explained in introduction.
Comment 5: What is the size of the treatment plots? Is there a replication of treatments?

Response 5: The size of the treatment plots was 30 m2, Yes, the treatments were replicated thrice and its mentioned in the section 2.2 Treatment description.

Comment 6: What depth were the soil samples taken?

Response 6: The soil samples were taken from 0-15 cm depth

Comment 7: Did you only perform the laboratory analysis of a soil sample per treatment (a mixture of 5 samples)? You only have one value per treatment! How is it possible to apply the mean comparison test? You should explain the methodological procedures very well!

Response 7: Yes, the laboratory analysis was performed on soil samples per treatment and from each treatment analysis was done on soil samples in three replications. The values given in tables are means of three replications. That’s why it was possible to apply mean comparison test.

Comment 8: Line 97-98 - “From each plot 5 samples were collected and they were mixed to get a representative sample of soil” - In this case, does Plot meaning treatment?

Response 8: The plot means treatment only.

Comment 9: Nutrient use efficiency: Explain the information obtained from the parameters evaluated in this item. For what purpose was each of them evaluated?

Response 9: The information obtained from parameters under nutrient use efficiency is given in results and discussion section. In materials and methods, we have just given the formulae to evaluate these parameters. The reasons behind evaluating them are explained in introduction.

Comment 10: Table 1 - The effects of treatments on soil pH and EC, despite undergoing small variations, are not significant, so there is not much to say. The OC increased significantly, but the values remain very low. Despite this, they bring benefits to the availability of nutrients for the plants. This fact should be highlighted in the text.

Response 10: The increase in OC content due to combined application of manures and fertilizers may be because of better crop growth accompanied by higher root biomass generation and higher return of leftover plant residues (L 350-352).

Comment 11: You should improve the conclusion

Response 11: Conclusions are reformulated.

Reviewer 2 Report

The submitted paper studied the effects of manures and chemical fertilizer addition on nutrient soil status and nutrient use efficiency in rice-wheat cultivation. The experiment is well designed, the results clearly presented and the discussion well written. All of it leads to clear conclusions.

I have some comments about parts that should be reformulated. 

In the introduction, there is a sentence "The chemical fertilizers lack in secondary and micronutrients so their sole application is unable to maintain the long-term soil health and crop productivity and organic manures 65 being low in nutrient status do not give spectacular results." - It is not correct, they can contain micronutrients, and are often in higher amounts than in manure.

In the treatments description remove the @ sign and reformulate this part.

Check the whole paper for different font size (lines 148-154, 161-163, 357-362, 365-370).

Author Response

Response to Reviewer 2 Comments

Comment #1 (Pages 1-2, lines 42-45): "The aim of nutrient use to improve...management practices.", the two sentences were separated. However it looks like they should be merged.

Response 1: (Page 1 Line 38-41): Two sentences are merged and missing letters are also added.

Comment #2 (Page 2, line 80): "...75% RDF...", all the acronyms throughout the manuscript should be fully defined at the first mention.

Respoanse 2 : The full form of RDF is given after Table 1.

Comment 3# (Page 2, lines 79-84): "The experiment comprised...strawcompost (T15).", it could be better adding a Table to summarize all the experiments and related experimental conditions. As it is, the description is not clear. A Table could be helpful for more clarity.  

Response 3: Table 1 and Table 2 are added to describe details of various treatment combinations and chemical composition of manures respectively.

Comment #4 (Page 6, line 232): "The persual of data...", was it "perusal of data"?

Response 4: Word “persual” is replaced with “perusal”

Comment #5: An important lack of this manuscript is related to more information about reasons which led the Authors to select the involved material (manure and fertilizers). Why did they select those various typologies of manure? What results they would expect from each one of the involved material? All these perspectives should be deepened and discussed. Moreover, a further Table reporting the physical-chemical characteristics of all the material involved should be added to this study. 

Response 5: The reasons for including various manures are described in introduction (Page 2). The chemical composition of organic manures is provided in Table 3.

Reviewer 3 Report

The present work focuses on an integrated approach for improving soil nutrient status and nutrient use efficiencies by application of both manures and fertilizers. The paper is generally well organized and discussed. It could be further improved by few more revisions addition.

Comment #1 (Pages 1-2, lines 42-45): "The aim of nutrient use to improve...management practices.", the two sentences were separated. However it looks like they should be merged.

Comment #2 (Page 2, line 80): "...75% RDF...", all the acronyms throughout the manuscript should be fully defined at the first mention.

Comment 3# (Page 2, lines 79-84): "The experiment comprised...strawcompost (T15).", it could be better adding a Table to summarize all the experiments and related experimental conditions. As it is, the description is not clear. A Table could be helpful for more clarity.  

Comment #4 (Page 6, line 232): "The persual of data...", was it "perusal of data"?

Comment #5: An important lack of this manuscript is related to more information about reasons which led the Authors to select the involved material (manure and fertilizers). Why did they select those various typologies of manure? What results they would expect from each one of the involved material? All these perspectives should be deepened and discussed. Moreover, a further Table reporting the physical-chemical characteristics of all the material involved should be added to this study. 

Author Response

Response to Reviewer 3 Comments

 Comment 1: Nutrient use efficiencies is replaced with Nutrient use efficiency in the title and abstract

Response 1: Nutrients Use Efficiencies….. replaced by Nutrients Use Efficiency

Comment 2: 2.1. Experimental site and soil characteristics is changed to Experimental site. All the characteristics of experimental soil are given in Table 3, 4 and 5 as initial status.

Response 2: line 73. 2.1. Experimental site and soil characteristics. At this session there are no soil characteristics presented, not even a reference of the soil. Also, a minimal characterization of the area and climate is needed to be add.

Comment 3 (Linne 153-154): The sentence “The application of organic manures and fertilizers in combination decreased the pH values at a much faster rate than sole application of organic manures” is modified to “The combined application of organic manures and fertilizers resulted in greater decrease in soil pH than sole application of organic manures.”

Rsponse 3: line 147. The application of organic manures and fertilizers in combination decreased the pH values at a much faster rate than sole application of organic manures. This statement is not necessary because the differences are absolutely insignificant.

Comment 4 (Line 342): There is a small change in pH with the addition of organic maures and the results are non-significant. So the word “slight” is added in the sentence.

Response 4: line 354. The decline in pH in manured plots Again, the difference between control and the lowest pH value is less than 0.3 unit. Can we consider this a huge drop of pH? Or could we even take it into consideration? I would say no. So, either you reconsider the statement or you add the slight decline

Comment 5: Conclusions is reformulated.

Response 5: I find the conclusions being very general. Here the best results it should be emphasized.

Reviewer 4 Report

See the documents attached

Author Response

Response to Reviewer 4 Comments

Comment 1: Nutrients Use Efficiencies….. replaced by Nutrients Use Efficiency

Response 1: Nutrient use efficiencies is replaced with Nutrient use efficiency in the title and abstract

Comment 2: line 73. 2.1. Experimental site and soil characteristics. At this session there are no soil characteristics presented, not even a reference of the soil. Also, a minimal characterization of the area and climate is needed to be add.

Response 2: 2.1. Experimental site and soil characteristics is changed to Experimental site. All the characteristics of experimental soil are given in Table 3, 4 and 5 as initial status.

Comment 3 : Line 147. The application of organic manures and fertilizers in combination decreased the pH values at a much faster rate than sole application of organic manures. This statement is not necessary because the differences are absolutely insignificant

Response 3: The sentence “The application of organic manures and fertilizers in combination decreased the pH values at a much faster rate than sole application of organic manures” is modified to “The combined application of organic manures and fertilizers resulted in greater decrease in soil pH than sole application of organic manures.”

Comment 4 : Line 354. The decline in pH in manured plots Again, the difference between control and the lowest pH value is less than 0.3 unit. Can we consider this a huge drop of pH? Or could we even take it into consideration? I would say no. So, either you reconsider the statement or you add the slight decline…

Response 4: There is a small change in pH with the addition of organic manures and the results are non-significant. So the word “slight” is added in the sentence.

Comment 5: I find the conclusions being very general. Here the best results it should be emphasized.

Response 5: Conclusions are reformulated.

Comment 6: I strongly recommend article to be revised from language point of view.

Response 6: The possible changes have been made.

Round 2

Reviewer 1 Report

The article is good and present important results.

The authors should put the depth of soil layer where they collected the soil samples.

Author Response

Reviewer 1

Comment 1: The title is too long.

Response 1: The title is already modified

Material and Methods

Comment 2: 2.1. Experimental site and soil characteristics - Despite the title, this item makes no reference to the soil characteristics.

Response 2: 2.1. Experimental site and soil characteristics are mentioned in the manuscript.

Comment 3: You should make a brief characterization of the soils, climate and geology of the experimental area.

Response 3: All the characteristics of experimental soil are given in Table 3, 4 and 5 as initial status and some additions are made in section 2.1 giving a brief characterization of climate and geology of the area.

Comment 4: It is necessary to make a clearer description of the treatments. What does the acronym RDF mean? Why did you choose these treatments? What factors were behind your choice?

Response 4: The acronym RDF means recommended dose of fertilizers. The reasons behind choosing these treatments are already explained in introduction.

Comment 5: What is the size of the treatment plots? Is there a replication of treatments?

Response 5: The size of the treatment plots was 30 m2, yes, the treatments were replicated thrice and its mentioned in the section 2.2 Treatment description.

Comment 6: What depth were the soil samples taken?

Response 6: The soil samples were taken from 0-15 cm depth

Comment 7: Did you only perform the laboratory analysis of a soil sample per treatment (a mixture of 5 samples)? You only have one value per treatment! How is it possible to apply the mean comparison test? You should explain the methodological procedures very well!

Response 7: Yes, the laboratory analysis was performed on soil samples per treatment and from each treatment analysis was done on soil samples in three replications. The values given in tables are means of three replications. That’s why it was possible to apply mean comparison test.

 Comment 8: Line 97-98 - “From each plot 5 samples were collected and they were mixed to get a representative sample of soil” - In this case, does Plot meaning treatment?

Response 8: The plot means treatment only.

Comment 9: Nutrient use efficiency: Explain the information obtained from the parameters evaluated in this item. For what purpose was each of them evaluated?

Response 9: The information obtained from parameters under nutrient use efficiency is given in results and discussion section. In materials and methods, we have just given the formulae to evaluate these parameters. The reasons behind evaluating them are explained in introduction.

Comment 10: Table 1 - The effects of treatments on soil pH and EC, despite undergoing small variations, are not significant, so there is not much to say. The OC increased significantly, but the values remain very low. Despite this, they bring benefits to the availability of nutrients for the plants. This fact should be highlighted in the text.

Response 10: The increase in OC content due to combined application of manures and fertilizers may be because of better crop growth accompanied by higher root biomass generation and higher return of leftover plant residues (L 350-352).

Comment 11: You should improve the conclusion

Response 11: Conclusions are reformulated.
